# Moral Disengagement, Empathy, and Cybervictim’s Representation as Predictive Factors of Cyberbullying among Italian Adolescents

**DOI:** 10.3390/ijerph18031266

**Published:** 2021-01-31

**Authors:** Maria Lidia Mascia, Mirian Agus, Maria Assunta Zanetti, Maria Luisa Pedditzi, Dolores Rollo, Mirko Lasio, Maria Pietronilla Penna

**Affiliations:** 1Department of Pedagogy, Psychology, Philosophy, Faculty of Humanistic Studies, University of Cagliari, 09123 Cagliari, Italy; mirian.agus@unica.it (M.A.); pedditzi@unica.it (M.L.P.); mirko.lasio@unica.it (M.L.); penna@unica.it (M.P.P.); 2Department of Brain and Behavioral Sciences, University of Pavia, I-27100 Pavia, Italy; zanetti@unipv.it; 3Department of Medicine and Surgery, Faculty of Medicine, University of Parma, 43126 Parma, Italy; dolores.rollo@unipr.it

**Keywords:** moral disengagement, cyberbullying, empathy, cybervictim, adolescents

## Abstract

This study aimed to evaluate which aspects of moral disengagement (MD), empathy, and representations of the victim’s experience (VER) could be predictors of cyberbullying (CB). One hundred and eight-nine students (11–17 years old) completed 3 self-report questionnaires: An MD scale, an empathy scale, and a CB questionnaire. In relation to the personal experience of CB, four groups were identified: Victim, bully, bully/victim, and no experience with CB. The linear bivariate correlation analysis shows correlations between empathy and VER, between empathy and MD, and between MD and VER. A multinomial logistic regression identified which predictors could increase a subject’s probability of belonging to one of the four groups regarding the personal experience of CB (victim, bully, bully/victim, no experience). Findings highlighted that low cognitive empathy might increase the probability for a student to belong to the bullies’ group, rather than the victims’ group. Furthermore, low perception of the consequences of CB on the victim might increase the probability of belonging to the bully, bully/victim, and no experience groups. Then, a high score in the diffusion of responsibility was a significant predictor of belonging to the victim group rather than the no experience group. Results from this study confirm the need for preventive measures against CB, including the empowerment of cognitive empathy, decreasing the diffusion of responsibility, and increasing the awareness of the consequences of CB on the victim.

## 1. Introduction

Cyberbullying (CB) is an international phenomenon which is increasing globally, specifically in recent years, and which has dangerous impacts on children and adolescents [1]. CB is defined as the aggressive behavior performed through media and technological communication, such as the internet and smartphones [2,3]. Indeed, with the advancements of technology, bullying problems have started to go beyond the face-to-face setting [4]. This phenomenon might be related to the fact that for children and adolescents (who are growing up in contact with new technologies), the distinction between online life and offline life ois often very fleeting; they might have difficulty identifying the boundary between joking and offensive behavior [5].

In order to deepen and clarify these problems, it is important to evaluate the risk factors associated with different forms of bullying and CB. For example, some studies conducted in past decades have highlighted the relevance of the relationship between moral disengagement (MD) and CB [6,7,8,9]. Other works have underlined the link among empathy and CB; indeed a body of literature states that low empathy predicts higher levels of CB [10,11,12,13].

### 1.1. Moral Disengagement and Cyberbullying

Several recent studies in the literature have analyzed the relationship between MD and aggressive online behaviors [6,8,9]. MD is grounded in social cognitive theory [14,15,16] and reflects a self-regulatory process whereby aggression perpetrators disengage from the aggressive acts by changing their beliefs and evaluations of the immoral act. Bandura [16] argued that the process of MD might be related to the redefinition of a lesioning behavior as positive behavior by a specific moral explanation. MD is also referred as “an influence on Traditional Bullying and Cyberbullying cognitive process, by which a person justifies his/her harmful or aggressive behavior, by loosening his/her inner self-regulatory mechanisms (…) which usually keeps behavior, in line with personal standards” ([17] p. 81). Bandura et al. [16,18] described some specific practices relevant to MD:Moral justification, where the individual cognitively restructures harmful conduct as acceptable behavior;Euphemistic labeling, which refers to language being sanitized in order to make harmful conduct appear benign;Advantageous comparison, in which the individual compares harmful acts with more reprehensible activities, so that they are viewed as having minor consequences;Minimizing or misconstruing consequences, which occurs when the results of a harmful act are minimized, ignored, or distorted to relieve the perpetrator from feelings of self-condemnation;Displacement or diffusion of responsibility, in which the harm done by a group can be attributed to others’ behavior, thus disowning personal responsibility;Attribution of blame, in which victims are considered to have brought their suffering upon themselves;Dehumanization, in which victims are stripped of human qualities.

Other works have shown that bullying is positively associated with self-reported MD in adolescents [19,20,21] and in children [22,23]. While the relationship between MD and traditional bullying has been widely proven in literature (e.g., [24]), the relationship between MD and CB has more recently been the focus of a specific interest in literature [18,24,25,26,27]. According to some authors [18,24,28], the connection between MD and CB is less significant compared to traditional bullying. Walrave and Heirman [29] stated that adolescents who perpetuate CB also tend to minimize the impact of their behavior on others; the results obtained show an underestimation of CB’s problem. The reasons for this may be found in the scenario (reality versus web), in the mechanisms connected to anonymity [17] and in the distance between the cybervictim and the cyberbully [30,31]. This aspect can be associated with online disinhibition, which refers to a reduction of restrictions and inhibitions during the online interaction compared to face to face. The online scenario might let individuals neglect their moral code, with consequent variations in terms of moral rules [32]. The inability to observe the victim’s immediate reaction would actually allow the aggressor to minimize the impact of their negative behavior, making moral disengagement less necessary [21,30]. Pornari and Wood [17] published a study involving British teenagers, which investigated their engagement in terms of traditional bullying behaviors and media-based aggressions. They highlighted a positive correlation between global MD and CB, showing that only moral justification is considered a significant predictor of CB. Other interesting results can be found in the work of Sheri Bauman [30], who states that MD is not a predictor of cyberbullying but is instead a predictor of the choice of acting out behaviors. Therefore, through analyzing the studies that focus on the comparison between perpetration and victimization, both in the context of traditional bully conduct and of CB, it is observed that in the web scenario there is little “need” to resort to cognitive processes of self-absolution and justification compared to with traditional bullying [3,17,18,27,30,33]. However, the meta-analysis carried out by Gini et al. [6] does not confirm these relationships.

### 1.2. Empathy and Cyberbullying

Empathy seems to act through two different channels of a cognitive and affective nature. Through cognitive empathy, it is possible to understand the emotions that other people feel or their emotional context. Affective empathy, on the other hand, allows one to experience and share the emotional states and emotional contexts of other people [34,35]. In face-to-face interactions, cognitive and affective empathy are both involved in natural empathetic replies. Some authors [36,37] have emphasized that the paucity of social cues connected to nonverbal and paraverbal languages, when employed in the new mobile media communication, might block the activation of affective empathetic processes. Pornari and Wood [17] have underlined that in online scenarios, bullies may not have a clear understanding of the consequences of their actions on the victim. Thus, they would not assume responsibility and would not feel they were to blame, minimizing the damage or the severity of their actions. Schultze–Krumbholz et al. [38], after reviewing numerous studies on the topic, stated that affective and cognitive empathy serve as protective factors against CB.

### 1.3. Aims

Referring to the above-mentioned literature, the findings are not univocal concerning the relationships between MD, empathy, and CB. This study therefore aimed at further exploring the associations between these dimensions in order to evaluate which mechanisms of MD, empathy, and representations of the victim’s experience (VER) might predict the personal experience of CB, with reference to the individuals belonging to the bully, the victim, the bully/victim, and the uninvolved subjects’ groups.

In particular, we hypothesized:That a relationship between empathy and VER, between empathy and MD, and between MD and VER exists;That a low level of MD increases the probability of belonging to the cyberbully group;That distorted representations of cybervictim characteristics predict the probability of being a cyberbully;That a low level of empathy predicts the probability of being a cyberbully.

## 2. Materials and Methods

### 2.1. Participants and Study Design

The survey involved 189 students (42% males) aging from 11 to 17 years (M_age_ = 13.2, SD_age_ = 1.2). One hundred and seventy-four students provided complete responses in an anonymous structured questionnaire. All participants belong to the medium socioeconomic level. The present work is part of a wider project conducted by the research team on Italian schools. The study was approved by the institutional committees of the schools in which the survey was administered. Specifically, our participants were recruited through nonprobability sampling across secondary schools (scientific high schools) in rural areas of Sardinia (Italy). Participants were identified in the schools involved in the project; six schools were initially contacted, from which four schools joined the project on the basis of their availability (specifically, with two classes for each school). Two hundred and ten alumni were contacted through the intermediary action of the school teaching staff. Among these, 189 alumni received informed consent to participate from their parents. From the initial sample of 189 students, 174 subjects who had completed the questionnaire in all its parts were examined. The sampling procedures are depicted in Figure 1.

### 2.2. Measurements

The protocol was distributed in large groups in a paper-and-pencil format; the work session lasted about 30 min. It was organized in different sections that were completed in one work session.

The first part of the protocol assessed the demographic variables (age, gender, socio-economic level).

#### 2.2.1. Basic Empathy Scale

The basic empathy scale (BES) by Albiero, Matricardi, and Toso [39] is the Italian validation of the BES [34]. This scale is made up of two subscales concerning two different components of empathic responsiveness: The affective empathy (11 items; Cronbach’s alpha = 0.85) and cognitive empathy (9 items, Cronbach’s alpha = 0.73). Agreement with the statements was indicated on a 5-point Likert-type scale.

#### 2.2.2. Moral Disengagement

Caprara, Bandura et al. [16] developed an Italian scale for MD composed of 32 items (Cronbach’s alpha = 0.82), assessed on a 5-point Likert scale (from 1 = strongly disagree to 5 = strongly agree). By means of this scale, some mechanisms of MD could be measured: Advantageous comparison, dehumanization of the victim, attribution of blame diffusion of responsibility, distortion of consequences, displacement of responsibility, moral justification, and euphemistic labeling.

#### 2.2.3. Representations of the Victim’s Experience

In order to assess the VER, a questionnaire composed of 35 items was distributed among participants and was measured on a Likert scale ranging from 1 (not at all) to 5 (extremely). The questionnaire used was prepared ad hoc on the basis of the available literature on the subject, with reference to the main thematic areas which had been identified (CB perpetration and CB victimization in relation to personal and situational factors [32,33,40,41]). The items investigated three dimensions: Consequences of CB on the victim (17 items; Cronbach’s alpha = 0.87, such as, “According to you, would a victim of CB struggle (a) with problems at school? (b) with problems at home? or (c) with friends?”); victim’s reactions to CB actions (8 items; Cronbach’s alpha = 0.69, for example, “According to you, should the victim of CB (a) adopt transgressive behavior or (b) be a cyberbully himself/herself?” “In your opinion, is a victim of CB who adopts transgressive behavior a victim and also cyberbully?”); and perceived predisposing factors for victimization (10 items; Cronbach’s alpha = 0.77, for example “In your opinion, if a victim of CB does not react to provocations, is he/she a weak person?” “Is he/she different from the others?”). The questionnaire appeared to be a valid and reliable tool to use to study teenagers’ attitudes in educational contexts.

In this section of the survey, in order to define the adolescent’s involvement and experience with CB, a further question was proposed: "Have you ever experienced CB?” (a) “Yes, I have been the victim of CB”; (b) “Yes, I have been a cyberbully”; (c) “I have been both cyberbully and cybervictim”; (d) “No, I have never experienced cyberbullying.” On the base of the responses given by participants to this question, they were assigned to the following groups: Victim, cyberbully, bully/victim, no experience of cyberbullying.

#### 2.2.4. Statistical Analyses

Some univariate, bivariate, and multivariate analyses were carried out in order to assess the effect of the above-mentioned aspects (MD, empathy, and VER) on involvement in CB. More specifically, a multinomial logistic regression was carried out [42]. This was applied to predict nominal outcome variables, in which the log odds of the outcomes were defined as a linear combination of the predictor dimensions. This statistical analysis was useful to define the effect of each predictor dimension (that could be nominal and/or continuous) in the categorical outcome variable (in this work, the answer given to the question inquiring the adolescent’s involvement with CB experience was assessed according to four modalities: Victim, cyberbully, bully/victim, no experience) [43]. Furthermore, as a specific advantage, this statistical technique made no assumptions about the distributions of classes in their feature space. Data analyses were conducted using the software SPSS version 22.0 (Statistical Package for the Social Sciences, International Business Machines Corporation (IBM), New York, NY, USA).

## 3. Results

Table 1 illustrates the descriptive statistics for the variables inquiring gender, age, and personal involvement in CB.

Moreover, the linear bivariate correlations (Pearson’s r) between the examined dimensions were applied (Table 2). These findings were consistent with the literature, showing positive correlations between the scales of empathy and the perceived consequence of CB on the victim. Moreover, a positive correlation between the scales of empathy and perceived predisposing factors for victimization was underlined. The dimensions of empathy also showed a significant negative correlation with the scales of MD and with the victim’s reactions to CB. The scales of MD were strongly correlated with them; all of them showed significant positive correlations with the victim’s reactions to CB. The assessment of the VER showed a positive correlation with the victim’s reactions and the perceived predisposing factors for victimization (see Table 2).

Finally, we applied a multinomial logistic regression, using as a dependent variable the multinomial categorical variable investigating the student’s experience in relation to CB. This variable corresponded to the question proposed in the questionnaire and assessed by four modalities: Victim, bully, bully/victim, no experience. In this logistic regression, several predictors were included: Age, gender, and the scales related to empathy, MD, and the dimensions implied in CB.

This analysis allowed for identifying which predictors could increase subject’s probabilities to be included in one of the four groups of the dependent variable (victim, bully, bully/victim, no experience). Indeed, in the multinomial logistic regression model, one outcome group was used as a “reference group” (also defined as the base category, in this case “victim”), and the coefficients for all other outcome groups defined how the predictor variables were related to the probability of being in that outcome group versus the reference group (in our analysis, the logistic regression evaluated the probability of belonging to each subgroup, compared with the reference group of “victim”).

The inclusion of predictors in the multinomial logistic regression equation was carried out with the enter method. All assumptions for the application of this statistical analysis were met.

The model presented a Cox and Snell pseudo-R-square = 0.305, Nagelkerke = 0.347, with an overall percentage of correct prediction of 64.4% (likelihood ratio test’s chi square = 63.374; df = 45; *p* = 0.037; goodness-of-fit test chi square = 484.815; df = 474; *p* = 0.356) (Table 3).

Specifically, the study showed that low cognitive empathy increased the probability of a student belonging to the bully group rather than the victim group. Moreover, the study stressed that a low perception of the consequences of CB on the victim increased the probability of the subject belonging to the bully group, to the group of individuals having had both experiences (bully/victim), and to the group of individuals who have never experienced CB. Finally, a high score in diffusion of responsibility proved to be a significant predictor of group membership for individuals who had never experienced CB rather than for the victims (see Table 3).

## 4. Discussion

This study attempted to investigate which aspects of MD, empathy, and VER might be related to adolescent involvement in CB. The findings underlined a connection between MD and CB, specifically with the aspect of MD related to the diffusion of responsibility. Furthermore, an interesting effect of cognitive empathy was observed in the personal experience with CB. In addition, awareness of the consequences of CB on the victim’s life showed a significant relationship with the adolescent’s experience of CB.

These findings might be read in light of a comparison between traditional bullying and CB; indeed, CB presents contextual and implicit features linked to online anonymity and disinhibition. In the web scenario, the need for cyberbullies to defend their reputation is less important. Moreover, cyberbullies would feel that they were less responsible because of the lack of contact with their victims [21,31].

Regarding the web scenario, the role of MD highlighted in our findings might be related to the moral self-regulation in Bandura’s model [14], which requires people to be able to decode even nonverbal signals: These signals are rather difficult to detect or are absent in the online environment, preventing the cyberbully from being aware of the resulting damage. Furthermore, the absence of direct contact between the perpetrator and victim might lower the cyberbully’s emotional engagement regarding feelings of remorse. Like empathy, remorse is an indicator of an individual’s awareness of the negative consequences of harmful acts on the victim; it thus might play the role of mediator between moral standards and moral behavior [21]. In the literature, it is emphasized that many features of CB (e.g., the absence of direct contact, distance from the victim, lack of visibility, anonymity) activate mechanisms that appear to attenuate or inhibit empathy and feeling guilty about CB. This aspect might make it easier for the bully both to act immorally, by either applying or not applying cognitive strategies that would allow the subject to disassociate from consequent moral responsibility, and not to experience guilty feelings [3]. Compared to the moral dimension, the potential invisibility of the victim might be a condition for a lack of empathy, which would not be activated. In the web scenario, the lack of direct contact (such as in face-to-face bullying and a normative social context—peer and adult), allows the subject to act immorally, without any perception of the suffering the victim might experience [9,44,45]. As previous studies have already shown, the mechanism of distorting consequences has been identified among the predictors of CB [27]. This mechanism is typical both of bullies and bullies/victims, and it is associated with the lowest levels of cognitive empathy in cyberbullies compared to the other subjects involved in electronic bullying.

However, further analysis is needed of the diffusion of responsibilities among subjects who are not involved in the phenomenon of CB. Those aspects could be also present among subjects who are not interested in practicing CB but who are aware of the risks perpetrators take in doing it [18].

Educators, schools, and communities need to build a school culture and specific guidelines that can provide a positive framework to better educate individuals about the use of the internet and avoiding aggressive or violent behavior in cyberspace. It is essential to intervene as promptly as possible to make young people aware of these profoundly important aspects. Educational institutions and schools responsible for the protection of young people should pursue the objective of combating episodes of CB through the establishment of territorial networks aimed at the implementation of joint projects to strengthen the know-how acquired.

## 5. Conclusions

The results of this study confirm the need for preventive measures against CB, which could be focused on the variable predictors of this phenomenon. Informing students about the socio-cognitive components connected to an aggressive behavior can stimulate awareness and control over online aggressive behaviors. MD could be then considered an element that might support the relationship between the moral interiorized principles and the individual’s actions. In light of the above considerations, it is therefore fundamental to promote the dimensions which could hinder the growth of CB through adequate educational policies aimed at promoting awareness and responsibility [2].

There are some limitations to this study, among which we highlight the use of a non-probabilistic sampling related to the availability of scholastic institutions, teachers, parents and adolescents. In the ongoing development of the research, it might be useful to enlarge the assessment to include a wider geographical area in Italy and different age groups in the survey. In addition, it might be useful in the future to include other relevant dimensions, such as self-esteem, emotional intelligence, smartphone addiction, and online disinhibition in relation to CB [31].

In conclusion, this explorative study might be considered a first step in the assessment of dimensions that might contribute to better defining adolescents’ engagement in CB by identifying dimensions that are potentially useful in developing programs of primary prevention. 

## Figures and Tables

**Figure 1 ijerph-18-01266-f001:**
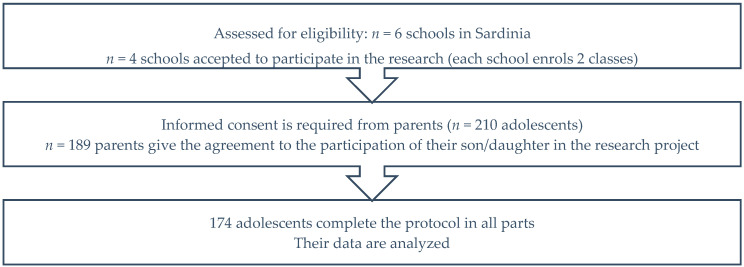
Flow of participants through each stage of research.

**Table 1 ijerph-18-01266-t001:** Descriptive statistics.

Variable		Frequency (%)
Gender	Males	85 (42%)
Age	Range	11–17
	Mean (±SD)	13.2 (±1.2)
Experience of cyberbullying	Victim	40 (23.0%)
	Cyberbully	9 (5.2%)
	Bully/victim	21 (12.1%)
	No experience of cyberbullying	104 (59.7%)

**Table 2 ijerph-18-01266-t002:** Descriptive statistics and Pearson’s correlation between the dimensions investigated.

					Dimension Number											
Dimension Number	Dimension	Range	Mean	SD	1	2	3	4	5	6	7	8	9	10	11	12
1	Cognitive empathy	2.00–5.00	3.68	0.72	1											
2	Affective empathy	1.55–5.00	3.44	0.67	0.445 **	1										
3	Moral justification	1.00–5.00	2.61	1.07	−0.07	−0.132	1									
4	Euphemistic labeling	1.00–5.00	1.93	0.93	−0.225 **	−0.208 **	0.515 **	1								
5	Advantageous comparison	1.00–5.00	1.68	0.86	−0.226 **	−0.116	0.398 **	0.674 **	1							
6	Displacement of responsibility	1.00–5.00	2.25	0.92	−0.190 *	−0.152 *	0.490 **	0.540 **	0.646 **	1						
7	Diffusion of responsibility	1.00–5.00	1.99	0.83	−0.261 **	−0.238 **	0.469 **	0.570 **	0.629 **	0.556 **	1					
8	Distortion of consequences	1.00–5.00	2.65	1.00	0.001	0.095	0.422 **	0.338 **	0.392 **	0.535 **	0.314 **	1				
9	Attribution of blame	1.00–5.00	2.12	0.85	−0.175 *	−0.076	0.491 **	0.584 **	0.590 **	0.545 **	0.532 **	0.370 **	1			
10	Dehumanization of victim	1.00–5.00	2.19	1.07	−0.089	−0.092	0.489 **	0.516 **	0.446 **	0.477 **	0.511 **	0.299 **	0.522 **	1		
11	Consequences of cyberbullying on the victim	1.06–4.94	3.28	0.71	0.293 **	0.176 *	0.053	−0.029	−0.116	−0.012	−0.058	0.058	−0.031	−0.006	1	
12	Victim’s reactions to cyberbullying actions	1.00–4.13	2.06	0.64	−0.153 *	−0.063	0.192 **	0.384 **	0.363 **	0.197 **	0.258 **	0.142	0.225 **	0.242 **	0.008	1
13	Perceived predisposing factors for victimization	1.00–5.00	2.95	0.79	0.404 **	0.271 **	−0.054	0.001	−0.117	−0.016	−0.136	0.032	0.006	0.021	0.452 **	−0.020

** *p* < 0.01 (2-tailed); * *p* < 0.05 (2-tailed).

**Table 3 ijerph-18-01266-t003:** Multinomial logistic regression coefficients defining the effect of the investigated dimensions on the personal experience of cyberbullying.

Dimensions	Cyberbully vs. Victim				Bully/Victim vs. Victim				No Experience vs. Victim			
	B	SE	Exp B (95% CI)	*p*	B	SE	Exp B (95% CI)	*p*	B	SE	Exp B (95% CI)	*p*
Age	0.051	0.408	1.052 (0.473–2.342)	0.901	−0.157	0.270	0.855 (0.503–1.453)	0.563	0.065	0.184	1.068 (0.744–1.532)	0.722
Gender	1.276	1.159	3.584 (0.370–0.3.732)	0.271	0.654	0.715	1.924 (0.474–7.813)	0.360	0.193	0.494	1.213 (0.460–3.195)	0.696
Cognitive empathy	−1.907	0.885	0.148 (0.026–0.841)	0.031 *	−0.905	0.542	0.405 (0.140–0.1170)	0.095	−0.655	0.369	0.519 (0.252–1.071)	0.076
Affective empathy	0.593	0.924	1.809 (0.296–11.056)	0.521	−0.186	0.547	0.830 (0.284–2.424)	0.734	−0.087	0.332	0.916 (0.478–1.756)	0.792
Moral justification	0.559	0.564	1.749 (0.579–5.277)	0.321	0.558	0.393	1.747 (0.808–3.777)	0.156	0.311	0.268	1.365 (0.808–2.308)	0.245
Euphemistic labeling	−1.312	0.845	0.269 (0.051–1.410)	0.120	−0.137	0.561	0.872 (0.290–2.621)	0.808	0.038	0.399	1.039 (0.476–2.270)	0.923
Advantageous comparison	0.771	0.737	2.163 (0.510–9.177)	0.296	0.125	0.584	1.134 (0.361–3.559)	0.830	0.017	0.426	1.017 (0.441–2.343)	0.969
Displacement responsibility	−0.128	0.597	0.880 (0.273–2.836)	0.830	−0.082	0.407	0.921 (0.415–2.045)	0.840	−0.311	0.314	0.733 (0.396–1.357)	0.323
Diffusion of responsibility	−0.154	0.849	0.858 (0.1602–4.529)	0.857	0.681	0.562	1.975 (0.656–5.946)	0.226	0.830	0.404	2.294 (1.039–5.066)	0.040 *
Distortion of consequences	0.829	0.573	2.292 (0.745–7.049)	0.148	0.264	0.403	1.302 (0.591–2.866)	0.512	0.134	0.271	1.143 (0.672–1.944)	0.622
Attribution of blame	0.122	0.745	1.129 (0.262–4.860)	0.870	−0.718	0.480	0.488 (0.190–1.250)	0.435	−0.345	0.341	0.708 (0.363–1.382)	0.311
Dehumanization of victim	0.065	0.531	1.067 (0.377–3.019)	0.903	0.559	0.344	1.749 (0.891–3.435)	0.104	−0.271	0.247	0.762 (0.470–1.237)	0.272
Consequences of cyberbullying on the victim	−1.846	0.821	0.158 (0.032–0.789)	0.025 *	−1.481	0.537	0.227(0.079–0.651)	0.006 **	−0.734	0.346	0.480(0.244–0.945)	0.034 *
Victim’s reactions to cyberbullying actions	0.476	0.773	1.610 (0.354–7.329)	0.538	−0.057	0.547	0.944 (0.323–2.759)	0.916	−0.171	0.374	0.842 (0.405–1.754)	0.647
Perceived predisposing factors for victimization	1.296	0.741	3.655 (0.855–15.617)	0.080	0.610	0.500	1.840 (0.691–4.902)	0.223	0.441	0.310	1.554 (0.846–2.854)	0.156

** *p* < 0.01 (2-tailed); * *p* < 0.05 (2-tailed); B = unstandardized coefficient; SE = Standard Error; Exp B= Odds ratio; 95% CI = 95% Confidence Interval for Exp B (lower-upper bound).

## Data Availability

The datasets for this study are available from corresponding author on reasonable request.

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
