# Peer review of "Moral Disengagement, Empathy, and Cybervictim’s Representation as Predictive Factors of Cyberbullying among Italian Adolescents"

_ijerph, 2021, doi:10.3390/ijerph18031266_

Round 1
Reviewer 1 Report
The present study addresses an important topic from an interesting perspective, linking moral disengagement, empathy and representations of the victim's experience as potential predictors of cyberbullying. However, there are some concerns that temper my enthusiasm for the manuscript in its current form.
- First, some sections of the manuscript are difficult to follow. This is particularly true for the Introduction section, drafted with sentences working as paragraphs, displaying some disconnected ideas. Relatedly, punctuation, writing and readability should be carefully revised thorough the manuscript.
- It would be helpful for the reader to have a more complete paragraph at the end of the Introduction section that really covers the objectives of the study (well structured), with related hypotheses.
- The Materials and Methods section can also be organized in a more comprehensive way, using subheadings (e.g., Participants and Procedures, Measures, Statistical Analyses). Using the label “About empathy” or “About MD” does not seem the proper way for a scientific paper. Also, the authors should include an internal consistency index (i.e., Cronbach’s alpha) for Empathy and MD measures.
- It is important to describe how the groups (bully, bully/victim, victim, no experience) were created from the “Representations of the victim’s experience”, including cut-off criteria. Related to this, the “Both” group could be labelled as Bully/victim, as observed in some previous research on the topic.
- Table 2 can be edited to increase readability, for example by changing to a horizontal format.
- In Table 3, for the Cyberbully vs Victim comparison only the B is displayed. Yet, for the remaining comparisons, the authors also include SE and Exp (B). I guess that it is because there are no variables predicting the inclusion in the bully group, a result that should be also further explained in the discussion section.
- From lines 207 to 213, I do not really understand if that information is really part of the Results section or the Discussion section.
- Overall, the Discussion needs to be revised, with a better connection between ideas and arguments. Theoretical and practical implications should be acknowledge, as well as the limitations of the study.
In sum, I consider that this manuscript needs more work in order to meet criteria to be published in this or any other journal. In this regard, since it covers an important idea, I think the authors could have another opportunity to improve it.
Author Response
Dear Editor and Referees,
Thank you for your letter Ref. Manuscript ID: ijerph-1050371, entitled “Moral Disengagement, Empathy and Cybervictim’s Representation as Predictive Factors of Cyberbullying among Italian Adolescents” (January 8, 2021) and for giving us the opportunity to review and resubmit the paper.
We are very grateful for your comments and suggestions and those of the reviewers. We are deeply appreciative of your careful reading. Detailed replies to your comments are enumerated below, with the list of modifications and integrations. We hope this revised version now satisfies the requirements for publication in your journal.
We are thus submitting this revised version of the paper. For clarity, new portions, added or modified in response to the referees’ comments, are highlighted in the manuscript (the tracked version of the manuscript is attached).
Thank you very much.
Best regards,
The Authors

Reviewer 2 Report
The sample that ended up being part of the study is well described, although there are certain doubts when considering that it may show high levels of significance and representativeness, especially if we take into account the large number of individuals in Italy who are going through the stage of adolescence and the fact that finally, among that large sample, only 189 adolescents between 11 and 17 years old ended up participating in the study.
The statistical treatment of the study data is described in a rather superficial and ambiguous way, so that the way in which they were treated and the information they managed to report to the study is not very clear.
The results of the a priori study appear to be congruent with the objectives set out in the article, although it is no less true that they are described in an excessively brief and superficial manner. Despite this, they conform to the methodological approaches established and planned in the study and are, for the most part, conveniently organized and written, at least from a chronological and statistical point of view.
The conclusions of the study are correctly stated, organized and described. However, despite this, they are unable to clearly delimit the scientific space through which the different empirical studies that, in the future, intend to continue with the trail or the path that they have stopped exploring will have to continue or run. the present study.
The references are quite current, despite the fact that in some cases they end up being older than ten years, and are reflected in the text of the article, scrupulously respecting the APA regulations in its seventh edition.
Author Response

(The authors gave the same response as above.)

Reviewer 3 Report
The manuscript describe the associations between empathy, MD, and cyberbullying in a small sample of Italian adolescents. The findings might contribute to develop effective intervention strategies for cyberbullying. However, there are several questions should be taken into consideration.
1.It is tough to read such a long introduction section. Authors should concisely and precisely introduce the background of the topic. For example, authors mentioned disinhibition many times in the manuscript, including in the introduction and discussion sections, but there had not any data on the disinhibition in the present study.
2.In the line 81, authors wrote, 'the cross-sectional survey by XXX', what does it refer to?
3.In the data analysis, authors did not describe how to select variables in the multivariate logistic regression model. According to the table 3, an enter method was used. Please verify it.
4. The results listed in the table 3 are insufficient. P values or 95%CI of the exp(B) should be listed.
5.Authors had better edit or rewrite the manuscript, especially the introdcution and discussion sections.
6. Authors had better describe the sampling process, i.e., how many students were contacted, how many of them refused etc.. A flow chart of the recruitment is advised.
Author Response

(The authors gave the same response as above.)

Round 2
Reviewer 1 Report
The authors have done a nice job in replying all the comments and, even more important, in integrating all suggestions in the manuscript, which has been considerably strengthened. I only have two minor suggestions:
- In regards Cronbach’s alpha, the values are reported with 2 decimals for empathy and moral disengagement scales, and with 3 decimals for the bullying/victimization scale. I would suggest, in order to keep the coherence, present all of them with 2 decimals.
- I appreciate that the authors included the questions used to assign participants to the bully, victim and bully/victim groups. Nevertheless, I still miss a brief explanation about the procedure used for that purpose, or at least a brief sentence explaining that based on the response provided to the question, participants were assigned to the intended groups.
Once this minor issues are solved, I consider that the manuscript would be ready for publication.
Author Response
Dear Editor and Referees,
Thank you for your letter Ref. Manuscript ID: ijerph-1050371 - Minor Revisions, entitled “Moral Disengagement, Empathy and Cybervictim’s Representation as Predictive Factors of Cyberbullying among Italian Adolescents” (January 24, 2021) and for giving us the opportunity to review and resubmit the paper.
We are very grateful for your comments and suggestions. We are deeply appreciative of your careful reading. Detailed replies to the comments are enumerated below.
We are thus submitting this revised version of the paper. For clarity, new portions, added or modified in response to the referees’ comments, are highlighted in the manuscript (the tracked version of the manuscript is attached).
Thank you very much.
Best regards,
The Authors

Reviewer 3 Report
All my concerns have been well addressed.Author Response
Dear Editor and Referees,
Thank you for your letter Ref. Manuscript ID: ijerph-1050371 - Minor Revisions, entitled “Moral Disengagement, Empathy and Cybervictim’s Representation as Predictive Factors of Cyberbullying among Italian Adolescents” (January 24, 2021) and for giving us the opportunity to review and resubmit the paper.
We are thus submitting this revised version of the paper. For clarity, new portions, added or modified in response to the referees’ comments, are highlighted in the manuscript (the tracked version of the manuscript is attached).
Thank you very much.
Best regards,
The Authors